# Allergic Disposition of IVF-Conceived Mice

**DOI:** 10.3390/ijms252312993

**Published:** 2024-12-03

**Authors:** Hamid Ahmadi, Zoltan Bognar, Timea Csabai-Tanics, Basil Nnaemeka Obodo, Julia Szekeres-Bartho

**Affiliations:** 1Department of Medical Biology, Medical School, University of Pecs, 7624 Pecs, Hungary; ahmadi.hamid@aok.pte.hu (H.A.); basil.obodon@gmail.com (B.N.O.); 2National Laboratory on Human Reproduction, University of Pecs, 7624 Pecs, Hungary; 3HUN-REN–PTE Human Reproduction Research Group, Hungarian Academy of Sciences, 1245 Budapest, Hungary; 4Department of Biological Sciences, Purdue University, West Lafayette, IN 47907, USA

**Keywords:** IVF, allergy, IL-4, IgE

## Abstract

With the increased utilization of assisted reproductive technology (ART), concerns about the potential health risks for ART-conceived babies have also been raised. Increased prevalences of allergic and metabolic diseases have been reported among ART offspring. This study aimed to evaluate the impact of IVF on the tendency to develop allergic responses following ovalbumin (OVA) sensitization in IVF-conceived mice. Mice were divided into four groups (non-OVA naturally conceived, OVA naturally conceived, non-OVA IVF-conceived, and OVA IVF-conceived). In the OVA groups, the mice were subjected to intraperitoneal and intranasal immunization with OVA. Two days after the final immunization, blood samples were taken, and the serum levels of IgE and IL-4 were detected by ELISA. The mice were sacrificed by cervical dislocation, their spleens and lungs were removed, and their weights were measured and recorded. Sensitization with OVA resulted in significantly increased concentrations of IL-4 and total IgE, as well as increased lung and spleen weights, among offspring from both natural and IVF conception. The concentrations of IgE and IL-4 and the lung and spleen weights in IVF-conceived mice were significantly higher compared to those in naturally conceived mice before and after sensitization with OVA. It is concluded that compared to naturally conceived mice, IVF-conceived mice exhibit a greater tendency to develop allergic responses against OVA.

## 1. Introduction

More than 40 years after the birth of the first in vitro fertilization (IVF)-conceived baby, assisted reproductive technology (ART) is now widely used to treat infertility-related problems. In parallel with increasing demands for ART, concerns about the adverse effects of ART on the health status of the resultant offspring have also been raised [1]. During ART processes, altered conditions, e.g., hormone treatment, gamete and embryo manipulation, and interfering in the natural selection of gametes, lead to genomic and epigenetic changes, which might result in increased risks of disorders throughout the later life of IVF-conceived babies [2,3,4].

Preimplantation embryos are highly sensitive to environmental cues. IVF is performed during the critical period of gamete and embryo development when the genome is undergoing drastic epigenetic remodeling, and any environmental alterations can easily affect normal developmental programming [2,3,4]. IVF-conceived babies might be at higher risk of developing diseases, e.g., high blood pressure and metabolic and epigenetic disorders [4,5,6,7], compared to their naturally conceived counterparts.

An increased risk of certain genomic imprinting disorders such as Beckwith–Weidemann syndrome, Angelman syndrome, Silver–Russel syndrome, and retinoblastoma has been reported in children conceived via ART [4]. The putative association between ART and genomic imprinting disorders indicates a synchrony between ART procedures and critical imprinting events that occur during gametogenesis, fertilization, and the early embryonic developmental stage [2]. It has been suggested that epigenetic modifications of imprinted genes that result from altered environmental conditions during early gametogenesis and embryogenesis in the preimplantation period may affect gene expression during later fetal development and result in altered phenotypic conditions in adulthood [4].

Immunological alterations in ART-conceived offspring have been investigated in animal models and in humans. Several studies indicated altered immune functions and increased rates of immune-related disorders in ART-conceived babies [8,9,10]. ART-conceived mice exhibited less effective immune responses to the BCG vaccine, and their immune response was skewed toward T-helper (Th) 2-dominated responses [9,10]. The increased prevalence rates of asthma, allergies, metabolic syndrome, and childhood diseases also indicate possible alteration of the immune system in IVF-conceived offspring [8,9,11].

Although the exact etiology is not completely understood, genetic–environmental interactions appear to play an important role in the development of allergy and asthma [12,13,14]. The biological effect of IgE is complex and is exerted by influencing the functions of several immune cells involved in the pathogenesis of allergic inflammation. The binding of IgE to FCεRI expressed on the surface of dendritic cells (DCs) enhances their antigen-presenting capability. It has also been confirmed that IgE captures allergens and facilitates their presentation to memory Th2 cells [15,16]. It is suggested that allergic diseases are mainly mediated by Th2 lymphocytes [16,17]. Once activated by allergens, specific Th2 cells can not only activate B cells to produce IgE but also recruit and activate basophils, mast cells, and eosinophils by secreting IL-4, IL-5, and IL-13. IL-4 is involved in IgE production and the pathogenesis of several aspects of allergic disease [16,17].

Several studies have assessed the risk of atopic diseases among ART-conceived babies. In a review study, it was concluded that the data on the risk of allergic diseases and asthma among ART-conceived offspring are inconsistent. Otherwise, some systematic reviews and meta-analyses, as well as population-based studies, have shown an increased rate of atopic disorders among ART-conceived offspring [14,18].

Many possible confounding factors could influence the results of epidemiological studies, and adverse outcomes have not been completely attributed to ART procedures. Since animal studies remove infertility as a potential confounding factor, such studies may provide useful information that could help in the authentication of human ART-related findings. Accordingly, the current study was conducted using IVF-conceived male mice to measure the serum levels of IgE and the weights of the lung and spleen following injection with ovalbumin (OVA), in comparison with naturally conceived counterparts.

## 2. Results

### 2.1. Characteristics of IVF and Outcomes

The characteristics and number of ovulated oocytes, the number of fertilized oocytes, the fertilization rate, the developmental stages, the number of transferred blastocysts, and the subsequent birth rates are presented in Table 1. The offspring were matched according to their weight, age, sex, and litter size.

### 2.2. Serum Levels of Total IgE in Naturally and IVF-Conceived Offspring

The concentrations of total IgE were measured in the sera of the subjects using a commercial ELISA kit. The basic IgE levels in IVF-conceived mice were slightly higher than those in naturally conceived ones. OVA sensitization in both naturally and IVF-conceived mice resulted in significantly increased serum IgE levels (*p* = 0.012 and *p* = 0.029, respectively) (Figure 1). The increase in IgE levels following OVA treatment was more pronounced in IVF-conceived than in naturally conceived mice. We also showed statistically significant increased serum levels of total IgE in OVA-IVF mice in comparison with OVA-NC counterparts (*p* = 0.004) (Figure 1).

### 2.3. Serum Levels of IL-4 in Naturally and IVF-Conceived Offspring

The concentrations of IL-4 were measured in the sera of the mice using a commercial ELISA kit. The basic IL-4 levels in IVF-conceived mice were significantly (*p* = 0.048) higher than those in naturally conceived ones. OVA sensitization in both naturally and IVF-conceived mice resulted in a significant increase in serum IL-4 levels when compared to their non-OVA-sensitized counterparts (*p* = 0.0148 and *p* = 0.0189) (Figure 2).

### 2.4. Lung Weights in Naturally and IVF-Conceived Offspring

The mice were sacrificed by cerebral dislocation, and their lungs were removed and weighed. Following OVA injection, the lung weights of both naturally and IVF-conceived animals were significantly higher compared to those of their non-injected counterparts (*p* = 0.034 and *p* = 0.013, respectively) (Figure 3). The results demonstrated significantly increased lung weights among N-IVF and OVA-IVF mice compared to their N-NC and OVA-NC counterparts (*p* = 0.0335 and *p* = 0.001, respectively) (Figure 3). Lung weights were normalized to body weights by multiplying the lung weight by 100 and then dividing the result by the total body weight. Our results demonstrated that sensitization with OVA resulted in increased lung index values in both IVF- and naturally conceived mice in comparison with their non-sensitized counterparts (0.88 vs. 0.68 and 0.76 vs. 0.625, respectively). The increase was more pronounced in IVF-conceived mice.

### 2.5. Spleen Weights in Naturally and IVF-Conceived Mice

The results of our study showed that OVA injection resulted in increased spleen weights in both naturally conceived and IVF-conceived offspring in comparison with their non-injected counterparts (*p* = 0.007 and *p* = 0.008) (Figure 4). We also demonstrated that the spleen weight in N-IVF mice was higher than that in N-NC ones (*p* = 0.013) (Figure 4). The spleen weight was significantly increased in OVA-IVF mice compared to their OVA-NC counterparts (*p* = 0.041) (Figure 4). The spleen weight was normalized by multiplying it by 100 and then dividing the result by the total body weight. Our results demonstrated that sensitization with OVA resulted in increased spleen index values in both IVF- and naturally conceived mice in comparison with their non-sensitized counterparts (0.526 vs. 0.49 and 0.671 vs. 0.599, respectively). The increase was more pronounced in IVF-conceived mice.

## 3. Discussion

IVF has been increasingly used to overcome infertility-related problems without any comprehensive assessment of its effects on the health status of resultant offspring. The underlying mechanisms of these possible health defects need to be clarified. Numerous studies suggest an association between ART and imprinting disorders [2,4,19].

Several studies have reported altered immune functions and an increased rate of immune-related diseases in ART-conceived babies [8,9,10]. Altered placental expressions of genes participating in immune responses, e.g., signal transducer and activator of transcription 4 (STAT4) and endoplasmic reticulum aminopeptidase 2 (ERAP2), have been confirmed in ART-treated women [19]. Disrupted placental development, including decreased utero-placental blood flow, reduced oxygen supply in the uterus, or abnormal maternal nutrient conditions, can alter the biological characteristics of trophoblast cells and result in abnormal placental function. Subsequently, the placenta may adapt by changing its functional protein expression or altering epigenetic modifications of placental gene expression to meet the necessities of fetal development [3,19].

Altered immune function and increased rates of immune-related diseases in ART-conceived babies have been reported [9,10]. It is suggested that epigenetic alteration can play a role in the pathogenesis of asthma. Epigenome-wide association studies of DNA methylation in blood related to asthma have indicated differential methylation in some specific gene regions [13]. ART may increase the incidence risk of allergic disease through direct epigenetic alteration. Based on these findings, we challenged naturally and IVF-conceived mice with OVA to evaluate the tendency of developing allergic reactions in IVF-conceived mice in comparison with naturally conceived counterparts.

In the present study, the sensitization of mice from both natural and IVF conception resulted in allergic reactions represented by higher serum levels of IgE and IL-4, as well as increased weights of the spleen and lungs. It has been demonstrated that OVA can induce IgE-mediated splenocyte proliferation, as well as Th-2 dominant immune responses [20]. It has also been suggested that OVA-sensitized mice develop a stronger inflammatory reaction in their airways [21].

Optimal immune hemostasis plays an essential role in supporting and stabilizing an individual’s immune responses [22]. IgE has been demonstrated to play an important role in type I immediate allergic responses [23]. The allergic asthmatic reaction to an allergen is linked to IgE-mediated mast cell activation that results in the accumulation of Th2 cells and eosinophils in the airway. By secreting cytokines such as IL-4, IL-5, and IL-13, Th2 cells have been linked to asthmatic conditions and inflammatory cell activation [24]. IL-4 plays essential roles in inducing IgE secretion by plasma cells and upregulated expression of FcɛRI and MHC class II molecules on basophils, mast cells, monocytes, macrophages, and B cells [24]. IL-4 also promotes the migration of Th2 cells and eosinophils to inflamed sites, induces goblet cell hyperplasia, and triggers airway hyperresponsiveness and mucus hypersecretion [25]. ART-conceived children show an increased risk of hospitalization for respiratory tract infections during early childhood [26]. A cohort study showed that asthma, wheezing, and the use of anti-asthmatic medications are more frequent among children born to subfertile couples using ART, in comparison with children born to healthy couples [8]. In addition, our previous study showed that compared to naturally conceived mice, ART-conceived counterparts exhibit less efficient immune responses against BCG vaccines through the further promotion of humoral and inflammatory-related immune response characteristics [9].

Consistent with the presented findings, our results showed slightly increased serum levels of IgE in non-sensitized IVF-conceived mice in comparison with their naturally conceived counterparts. Following sensitization with OVA, IVF-conceived mice had significantly higher serum levels of IgE in comparison with naturally conceived ones. From these data, we can infer that mice conceived via IVF tend to have elevated serum IgE levels and might face an increased risk of allergic disease, particularly following exposure to allergens.

The spleen plays a critical role as a secondary lymphatic organ where the antigen-specific T-cell responses and antibody secretion by B cells take place, resulting in systemic immune responses [27]. The frequencies of splenic mast cells increase under inflammatory conditions such as chronic allergic dermatitis and food allergies (FAs) [28]. In a Th2 response-mediated FA model induced by OVA, a significant expansion of mast cells was observed in the spleen [21]. Asthma is an inflammatory disease of the airways involving various inflammatory and immune cells such as eosinophils, mast cells, neutrophils, and lymphocytes. An increased influx of inflammatory cells into airways following OVA injection has been demonstrated in mouse models [21]. In an experimental study, it was shown that male mice conceived via IVF had larger spleens and livers compared to their naturally conceived counterparts [29]. In line with these results, we demonstrated increased spleen and lung weights in male mice conceived via IVF compared to naturally conceived ones before injecting them with OVA. We also showed a greater increase in spleen and lung weights in offspring produced by IVF than in those produced by natural conception following allergy induction with OVA.

Although the defined cause is not completely indicated, genetic and environmental factors may be involved in the development of asthma and allergy [30]. ART may increase the risk of developing atopic disorders in offspring through direct epigenetic alterations [14]. Infertility and ART are stressful events for couples, as evidenced by increased cortisol levels and anxiety symptoms during pregnancy [31]. Maternal stress could enhance cortisol levels during pregnancy, and the transplacental passage of cortisol could alter the development of the hypothalamic–pituitary–adrenal axis in the fetus, resulting in an altered stress response later in life [31]. Prenatal maternal stress affects asthma and allergy development through the incomplete development of the respiratory tract and immune dysregulation in the offspring [32]. It has also been demonstrated that maternal stress may result in an altered composition of the offspring microbiota. These alterations may adjust the development of the immune system and increase the risk of asthma and allergic disorders in the offspring [32].

A collective comparison of OVA-induced inflammatory responses across the four groups revealed that IVF results in a greater tendency to develop allergic reactions in offspring, as shown by increased serum levels of IgE and IL-4 and enhanced lung and spleen weights among IVF-conceived mice in comparison with their naturally conceived counterparts.

## 4. Materials and Methods

### 4.1. Animals

Six-week-old male and female CBA and B6 mice were purchased from Charles River (Erkrath, Germany). Oocytes and sperm were obtained from CBA mice, while B6 males and females were used as vasectomized mice and surrogate mothers, respectively. Animals were kept in a light- and temperature-controlled room (22 °C, 14 h light/10 h dark, 50% ± 10% humidity) in pathogen-free conditions and fed ad libitum with a standard diet. This study was approved by the Animal Health Committee of Baranya County, Hungary.

### 4.2. Preparation of Gametes

CBA female mice were injected with 7.5 IU of pregnant mare serum gonadotropin (PMSG) (Meriofert, Rowe, Hackensack, NJ, USA), followed by 7.5 IU of human chorionic gonadotropin (hCG) (Ferring, Budapest, Hungary) 48 h later. Cumulus–oocyte complexes (COCs) were harvested from the oviducts 12–15 h following hCG injection by oviduct recession after cervical dislocation. The COCs were treated with 1% hyaluronidase (Vitrolife, Gothenburg, Sweeden) to remove the cumulus cells. The oocytes were incubated for 30 min before IVF. Immediately after cervical dislocation, sperm were obtained from the cauda epididymis of mature CBA male mice. The harvested sperm cells were transferred into 100 mL of potassium simplex optimization medium (KSOM) (Merck, Rahway, NJ, USA) and then incubated at 37 °C for 1.5 h to allow capacitation.

### 4.3. In Vitro Fertilization (IVF) and Natural Conception

To produce naturally conceived offspring, two female mice were mated with one male mouse. The next morning, they were checked for the presence of vaginal plugs, and those with vaginal plugs were kept individually in separate cages until the day of delivery. To produce IVF-conceived mice, IVF was performed as previously described [10] with minor modifications. Briefly, 1 × 10^5^ capacitated sperm cells were added to oocytes in KSOM and incubated at 37 °C for 6 h. Subsequently, the fertilized zygotes were washed serially in M2 medium (Merck, Rahway, NJ, USA) to remove the excess spermatozoa, debris, and cumulus cells and transferred into an equilibrated drop of KSOM, covered with paraffin oil. The embryos were cultured at 37 °C with 5% CO_2_ until they reached the blastocyst stage.

### 4.4. Pseudo-Pregnant Mice and Embryo Transfer

To prepare pseudo-pregnant mothers, female and vasectomized male B6 mice were mated overnight. The next morning, female mice were assessed for the presence of a vaginal plug. Females with vaginal plugs were considered pseudo-pregnant. Then, 7–10 embryos at the blastocyst stage were transferred into each side of the uterus. Seventeen days following embryo transfer, the pups were delivered naturally.

### 4.5. Experimental Groups

Animals were divided into 4 groups: naturally conceived non-OVA-treated (NC) (n = 8), naturally conceived OVA-sensitized (OVA-NC) (n = 13), IVF-conceived non-OVA-treated (IVF) (n = 8), and IVF-conceived OVA-sensitized (OVA-IVF) (n = 13). The total body weights of the mice were measured and recorded before sacrifice. To produce an OVA-induced inflammation model, male mice at 8 weeks were injected intraperitoneally on days 1 and 8 with 20 µg of OVA (Sigma, St. Louis, MO, USA) mixed with 1 mg of aluminum hydroxide (Sigma-Aldrich, Seoul, Republic of Korea) dissolved in a total volume of 200 µL of saline. On days 15, 16, and 17 after the initial sensitization with OVA, the mice were injected intranasally with 20 µg of OVA dissolved in 40 µL of saline. In the NC and IVF groups, mice were injected with an equal volume of normal saline. Intranasal administration was performed by pipetting saline or 20 µg of OVA dissolved in 40 µL saline onto the outer edge of the noses of the mice. Forty-eight hours after the last challenge with OVA, the mice were anesthetized, and blood samples were taken by cardiac puncture. Approximately 1 mL of blood was collected from each mouse. The blood samples were centrifuged, and the sera were separated and stored at −80 °C until use. The organ collection for analysis was performed on sacrificed animals.

### 4.6. Measurement of Organ Weights

Mice were matched based on their weight and sacrificed by cervical dislocation at 10 weeks of age; the viscera and the abdominal and thoracic cavities were investigated carefully. The spleen and lungs were removed; their weights were measured using a scale and recorded.

### 4.7. Evaluation of Total IgE and IL-4 Concentrations

The total IgE and IL-4 concentrations in serum were measured by ELISA kits (both Invitrogen, Waltham, MA, USA, cat.no: EMIGHE and BMS613, respectively) according to the manufacturer’s instructions. The optical density was determined by measuring the absorbance at 450 nm using an ELISA reader (Absorbance 96, byonoy GmbH, Hamburg, Germany).

### 4.8. Statistical Analysis

The results obtained from independent experiments are presented as the mean ± SD. The ANOVA statistical test was used to compare the mean difference in the studied markers among the four groups. *p* ≤ 0.05 was considered statistically significant.

## 5. Conclusion

According to the limited results of this study, it is suggested that stress applied during the preimplantation period in the IVF procedure may exert adverse effects on the long-term health of the offspring. To the best of our knowledge, this is the first experimental study to examine whether IVF-conceived mice have a greater tendency to develop allergies following sensitization with OVA. As mentioned earlier, data about the health status of ART-conceived babies are controversial, so continuous follow-up and the use of animal models for more laboratory investigations are of utmost importance. Moreover, since the procedures and culture media used in mouse and human IVF are different, it is not possible to completely extrapolate the results from mice to humans.

## Figures and Tables

**Figure 1 ijms-25-12993-f001:**
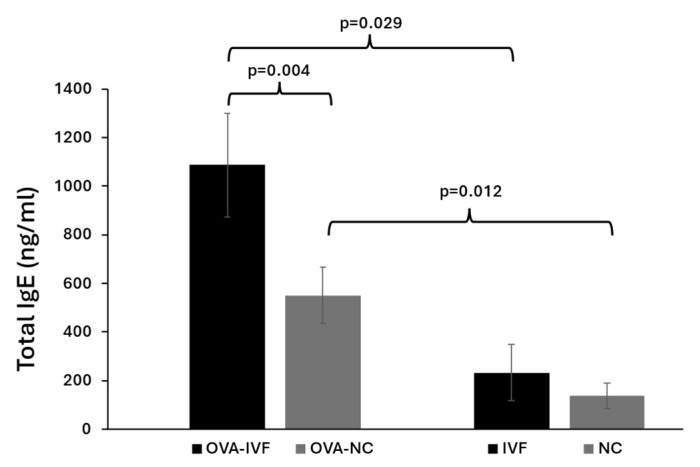
Serum levels of total IgE in naturally conceived non-OVA-treated (NC) (n = 8), naturally conceived OVA-sensitized (OVA-NC) (n = 13), IVF-conceived non-OVA-treated (IVF) (n = 8), and IVF-conceived OVA-sensitized (OVA-IVF) (n = 13) mice. The bars represent the mean ±SD of 8, 13, 8, and 13 independent experiments, respectively. *p* ≤ 0.05 was considered statistically significant.

**Figure 2 ijms-25-12993-f002:**
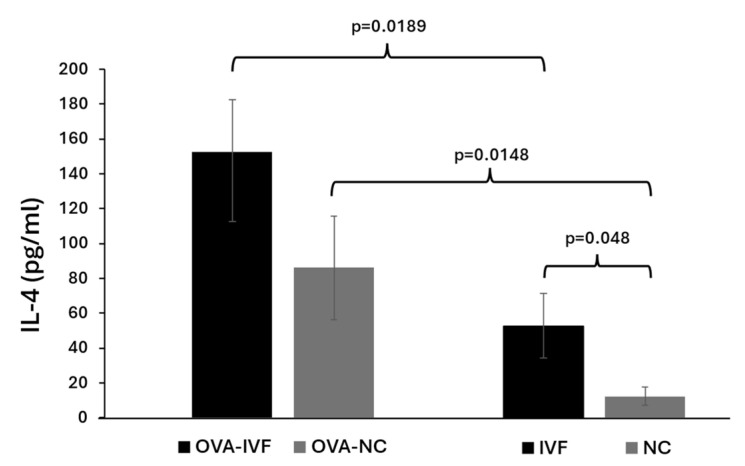
Serum levels of IL-4 in naturally conceived non-OVA-treated (NC) (n = 8), naturally conceived OVA-sensitized (OVA-NC) (n = 13), IVF-conceived non-OVA-treated (IVF) (n = 8), and IVF-conceived OVA-sensitized (OVA-IVF) (n = 13) mice. The bars represent the mean ±SD of 8, 13, 8, and 13 independent experiments, respectively. *p* ≤ 0.05 was considered statistically significant.

**Figure 3 ijms-25-12993-f003:**
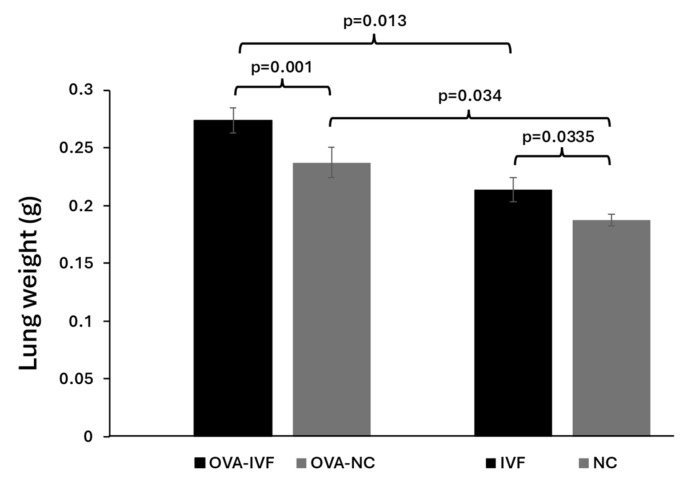
The weights of lungs of naturally conceived non-OVA-treated (NC) (n = 8), naturally conceived OVA-sensitized (OVA-NC) (n = 13), IVF-conceived non-OVA-treated (IVF) (n = 8), and IVF-conceived OVA-sensitized (OVA-IVF) (n = 13) mice. The bars represent the mean ±SD of 8, 13, 8, and 13 independent experiments, respectively. *p* ≤ 0.05 was considered statistically significant.

**Figure 4 ijms-25-12993-f004:**
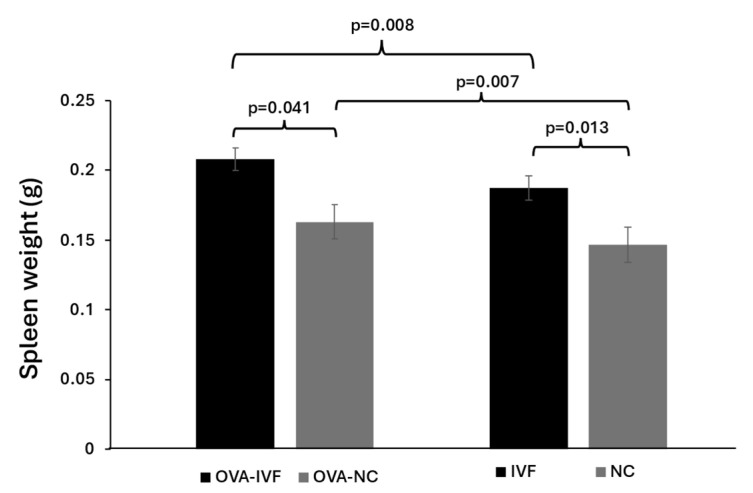
The weights of spleens of naturally conceived non-OVA-treated (NC) (n = 8), naturally conceived OVA-sensitized (OVA-NC) (n = 13), IVF-conceived non-OVA-treated (IVF) (n = 8), and IVF-conceived OVA-sensitized (OVA-IVF) (n = 13) mice. The bars represent the mean ±SD of 8, 13, 8, and 13 independent experiments, respectively. *p* ≤ 0.05 was considered statistically significant.

**Table 1 ijms-25-12993-t001:** The numbers of oocytes and two-cell, four-cell, and blastocyst-stage embryos; the numbers of transferred blastocysts; the numbers of offspring; and IVF and birth rates.

Cycle	Ovulated Oocytes	Two-Cell Stage	IVF Rate (%)	Four-Cell Stage	Blastocysts	Transferred Blastocysts	No. of Recipients	Offspring	Birth Rate (%)
1	35	15	42.85	12	8	8	1	4	50
2	37	26	70.27	26	24	22	2	4	18.8
3	40	30	75	26	25	12	1	3	25
4	24	18	75	18	18	15	1	3	20
5	20	16	80	14	14	14	1	3	21.42
6	50	32	64	30	24	24	2	6	25
7	37	26	70	26	24	10	1	2	20

The offspring were selected and matched with their naturally conceived counterparts based on their sex, age, weight, litter size, and health status.

## Data Availability

The original contributions presented in this study are included in the article. Further inquiries can be directed to the corresponding authors.

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
