# Peer review of "Allergic Disposition of IVF-Conceived Mice"

_ijms, 2024, doi:10.3390/ijms252312993_

Round 1

Reviewer 1 Report

Comments and Suggestions for Authors

Dear authors,

I read with interest your manuscript. The aim of the study is relevant as it correlates with human IVF concerns. However, as the authors appropriately mention in their conclusions, results obtained in their experiments with mice can not be translated to human IVF straightforward. Thus, the present study represents a contribution to further research on human IVF and its putative harmful effects on individual health issues. 

Kind regards

Author Response

I read with interest your manuscript. The aim of the study is relevant as it correlates with human IVF concerns. However, as the authors appropriately mention in their conclusions, results obtained in their experiments with mice can not be translated to human IVF straightforward. Thus, the present study represents a contribution to further research on human IVF and its putative harmful effects on individual health issues. 

Kind regards

Responses to reviewer 1

Dear Reviewer

Thank you for your positive feedback on our manuscript “Allergic disposition of IVF conceived mice.”  We appreciate your time and efforts in reviewing our manuscript.

Reviewer 2 Report

Comments and Suggestions for Authors

The work titled “Allergic disposition of IVF conceived mice ” studies how exposure to ovalbumin to naturally conceived mice as opposed to IVF generated mice affects spleen and lung weight and IgE and IL4 levels at age 10 weeks.  Four groups (2 non exposed and 2 exposed to OVA) were generated.

The authors found that:

1.        Sensitization with OVA results in significantly increased concentrations of IL-4, toltal IgE as well as increased lung and spleen weights among offsprings from both natural and IVF conceptions.

2.       The concentrations of IgE, IL-4 and the of the lung and spleen weights in IVF-conceived mice were significantly higher compared to naturally conceived ones before and after sensitization with OVA.

The authors conclude  that compared to naturally conceived mice, IVF-conceived mice exhibit a higher tendency to develop allergic responses against OVA.

The overall subject area is important given the very large number of IVF conceived children. It is also valuable to understand if allergic reactions occur, given that clinical evidence is limited.  The work is valuable, however there are few critical concerns that need to be addressed

1.       First,  litter size in both natural and IVF mice with birth weight and weekly growth curve to sacrifice needs to be reported. Table 1 needs to add how many recipients were used for each IVF experiments.  Litter size could be a major confounding factor.  Weight of organs should be normalized to total body weight at sacrifice

2.       Given known sexual dimorphic effect of IVF on outcome, results need to be divided for males and females. Reference 29 reports that only males had larger spleen

Minor points

3.       Line 265: MC please spell

4.       Line 285: pro-oxidant – antioxidant: if the results were not statistically different, the statement should be removed.

Reviewer 3 Report

Comments and Suggestions for Authors

The manuscript is clearly written, and the main goal is thoroughly explained. Although there are a lot of publications, showing the effects of in vitro fertilization on allergic processes, asthma, pulmonary hypertension already in humans research, and not only on an animal model.  Xu et al. show that children born after fresh embryo transfer exhibit an increased risk of immune dysfunction in childhood, manifested by elevated IL-4 serum levels and decreased IFN-É£/IL-4 ratio (Xu X., Wu H., Bian Y., Cui L., Man Y., Wang Z., Zhang X., Zhang C., Geng L. The altered immunological status of children conceived by assisted reproductive technology. Reprod. Biol. Endocrinol. 2021;19:171. doi: 10.1186/s12958-021-00858-2). In addition, here the authors did not explain the mechanisms, but only state the fact. Perhaps, based on recent research, the authors present a gap in the literature that they try to fill with their analysis, but these results are no big novelty.  This study aimed to evaluate the impact of IVF on the tendency to develop allergic responses fol- 12 lowing ovalbumin (OVA) sensitization in offsprings from IVF conceived mice. The methodology is well written and includes many details that are important for its replicability and/or reproducibility.

However, some details should be provided.

1.       The authors mentioned that the ELISA kits were purchased from Invitrogen-USA, but there are no catalogue numbers. This information is important, especially considering the number of ELISA kits available.

2.       What microplate reader was used in this study?

3.       The T-student test is used only for pairwise comparisons. Why was it used in the study when the authors have 4 groups? The use of the t-student test in this case is inappropriate and generates false conclusions.

4.       Tables and figures are clearly presented.

5.       The authors interpreted the data obtained in this study and compared them with the analysis of other research groups. The authors in the manuscript relied about 66% on literature reports older than 5 years.

6.       The manuscript is clearly written and focused on the topic being discussed. Individual parts of the work are well arranged, and subsequent analyses guide the reader through the researchers' thinking.

Round 2

Reviewer 2 Report

Comments and Suggestions for Authors

The resubmission has satisfied the majority of questions

However, few changes are needed:

1. The  litter size of  natural group at birth and during weaning needs to be added

2. The method section needs to explain that female mice were sacrificed: when? And how?

3. Litter size during lactation needs to be reported in both growth

4. The discussion needs to include a paragraph explaining that only male were tested and this is a limitation of the study

5. The discussion needs to include a paragraph discussing that different litter size can results in different phenotype and this is a limitation of the study

Author Response

Please, see attachment.
